# Anthocyanin of Black Highland Barley Alleviates H_2_O_2_-Induced Cardiomyocyte Injury and Myocardial Infarction via Activating the Phosphatase and Tensin Homolog/Phosphatidylinositol 3-Kinase/Protein Kinase B Pathway

**DOI:** 10.3390/foods13091417

**Published:** 2024-05-05

**Authors:** Zhendong Liu, Senbiao Shu, Simin Li, Pai Peng, Ying Zhang, Zhaohua Li, Wenhan Wang

**Affiliations:** 1Food Science College, Tibet Agriculture & Animal Husbandry University, Nyingchi 860000, China; liu304418091@126.com (Z.L.); shu19980724@126.com (S.S.); min201426@126.com (S.L.); 17852035465@163.com (P.P.); 18238451475@163.com (Y.Z.); 2Institute of Biophysics, Chinese Academy of Sciences, Beijing 100101, China; 3Institute of Edible Fungi, Shanghai Academy of Agricultural Sciences, Shanghai 201403, China

**Keywords:** cardiovascular disease, black barley, anthocyanin, oxidative stress, PTEN/PI3K/Akt pathway

## Abstract

Cardiovascular disease (CVD) represents a substantial global health challenge, with its impact on mortality and morbidity rates surpassing that of cancer. The present study was designed to explore the cardioprotective properties of anthocyanin (ACN), a compound derived from black barley, against oxidative stress-induced damage in myocardial cells and to uncover the molecular mechanisms at play. Utilizing both in vitro and in vivo experimental models, our findings indicate that ACN notably reduced cell damage caused by oxidative stress and effectively prevented apoptosis. High-throughput RNA sequencing analysis has shed light on the mechanism by which ACN achieves its antioxidative stress effects, implicating the PTEN-Akt signaling pathway. ACN was found to modulate PTEN expression levels, which in turn influences the Akt pathway, leading to a reduction in apoptotic processes. This novel insight lays the groundwork for the potential clinical utilization of ACN in the management of CVD. While this study has shed light on some of the functions of ACN, it is important to recognize that natural compounds often interact with multiple molecular targets and engage in intricate signaling cascades. Future research endeavors will concentrate on further elucidating the regulatory mechanisms by which ACN influences PTEN expression, with the goal of enhancing our comprehension and expanding the therapeutic potential of ACN in the treatment of cardiovascular conditions.

## 1. Introduction

Cardiovascular disease (CVD) is a prevalent and life-threatening condition worldwide, with mortality and morbidity rates exceeding those of cancer [1]. High levels of reactive oxygen species (ROS) induce myocardial cell damage, playing a significant role in the pathogenesis of many heart diseases, including cardiac hypertrophy, heart failure, myocardial infarction, and myocardial ischemia/reperfusion injury. Several antioxidants, such as vitamin C, vitamin E, and Coenzyme Q10, have been demonstrated to prevent and treat various forms of cardiovascular diseases caused by excessive oxidative stress and the generation of reactive oxygen species (ROS) [2]. Therefore, exploring effective antioxidants and their underlying mechanisms may provide opportunities for the treatment of cardiovascular diseases.

Anthocyanidin (ACN) is a water-soluble natural pigment that is widely present in plants. It exhibits powerful antioxidant properties [3]. Epidemiological studies have shown that the long-term consumption of foods rich in anthocyanins can effectively reduce the mortality rate of cardiovascular disease (CVD) [4]. Flavonoids including anthocyanins have a significant improvement effect on some CVD risk biomarkers (such as NO, inflammation, and endothelial dysfunction) [5,6,7,8]. Some studies have revealed the antioxidant effects of anthocyanins on the cardiovascular endothelium, but the mechanism by which they protect against oxidative stress-induced myocardial damage remains incompletely understood [9]. However, research also shows that this effect has a more complex molecular mechanism, including the regulation of gene expression, cell signal transduction, and miRNA expression [10]. Further investigation into the mechanisms of ACN against oxidative stress and oxidative stress-induced myocardial cell damage holds potential therapeutic significance for its application in cardiovascular disease treatment [11,12].

Highland barley (*Hordeum vulgare* L. var. *nudum Hook.f*) belongs to the genus *Hordeum* of the Poaceae family and is a variant of multi-rowed barley. It is also called naked barley, yuan barley, or rice barley. It is rich in nutrients and suitable for growing at high altitudes. It is an important food crop with ethnic and regional characteristics in China’s Qinghai–Tibet Plateau, with a cultivation history of about 3500 years. According to different colors, highland barley can be divided into white, blue, purple, and black highland barley [13]. As a precious germplasm resource, black highland barley is rich in various nutrients needed by the human body and contains more nutrients than other varieties of highland barley. Rich in anthocyanins, polyphenols, and other bioactive substances, it has strong anti-aging, antioxidant, and anti-inflammatory effects [14]. The extraction methods of black highland barley anthocyanins include solvent extraction, ultrasonic-assisted extraction, etc. The solvent extraction method can extract plant materials from various sources, and the extracted anthocyanin structure is relatively stable. Extraction solvents include ethanol, methanol, acetone, dimethyl sulfoxide, etc. [15]. The ultrasonic-assisted extraction method is based on the solvent extraction method and is supplemented by ultrasonic waves.

The use of H_2_O_2_ to induce cellular oxidative damage is a common method for establishing oxidative stress cell models. H_2_O_2_ easily penetrates the cell membrane and reacts with intracellular iron ions to generate highly reactive free radicals. This method is simple to use, and the model is stable and reliable, making it widely applied in numerous studies [16]. Cell transcriptome analysis is currently widely used to study the mechanism of action of drugs on cells [17]. Although there has been considerable progress in understanding the antioxidant properties of anthocyanins, their mechanism of action still requires further exploration.

Therefore, in this investigation, we utilized H_2_O_2_ to trigger apoptosis in H9c2 cells and performed surgical induction of myocardial infarction in rats to, respectively, establish in vitro and in vivo models. These models facilitated the assessment of various experimental indices and parameters, enabling us to delve into the protective role and underlying mechanisms of anthocyanins derived from black highland barley against cardiovascular diseases, particularly under conditions of oxidative stress.

## 2. Materials and Methods

### 2.1. Overview of Experimental Design and Procedures

The harvested black barley was first washed, crushed, and dried, and then soaked in acidic 70% ethanol (pH = 2) with ultrasound to extract the anthocyanins. Purified anthocyanins were identified using high-performance liquid chromatography–mass spectrometry (HPLC-MS) (Agilent 6545 Q-TOF LC/MS, Agilent Technologies, Inc., El Cajon, CA, USA). At the same time, the experiment established cardiomyocyte models and experimental animal models which were used to measure different experimental indicators and parameters including cell viability analysis, an apoptosis assay, the determination of biochemical indicators, a histopathological assay, and transcriptome information analysis when treated with anthocyanins (Figure 1).

### 2.2. Isolation, Purification, and Identification of Anthocyanins from Black Highland Barley

Black highland barley was provided by the Laboratory of Molecular Breeding of Tibetan Plateau Crops, which is the Key Laboratory of the Department of Education at Tibet Agricultural & Animal Husbandry University. After washing, crushing, and drying, the black highland barley powder was obtained and stored in a sealed bag at 4 °C. All other chemicals were of analytical grade.

A referenceable method for anthocyanin purification was applied [18]. Fifty grams of black highland barley powder was added to a solution of 70% ethanol (500 mL, pH = 2). The samples underwent ultrasound treatment for 4 s followed by a 4 s interval, repeated over a total duration of 40 min at 400 W. After sonication, the samples were left to stand for 12 h at 15 °C. Subsequently, the supernatant was obtained and concentrated through centrifugation at 4000 rpm for 30 min. The concentrated supernatant was then applied to the column until the macroporous resin (HP-2MGL) (H&E Co., Ltd., Beijing, China) reached adsorption saturation. The resin was rinsed with distilled water until the eluate was colorless. Subsequently, the sample was eluted with various concentrations of ethyl alcohol solution (10%, 50%, 75%, 95%). The eluate was collected, concentrated, and then lyophilized. Anthocyanins from black highland barley are denoted as ACN.

Anthocyanins were identified using high-performance liquid chromatography–mass spectrometry (HPLC-MS) [19]. HPLC analysis was conducted using an Agilent 1290 Infinity system equipped with an Alltima C18 reversed-phase column (250 × 4.6 mm, 5 μm) (Avantor, Radnor, PA, USA). The mobile phase consisted of a 2% formic acid solution and a 2% formic acid–acetonitrile solution. A 20 μL sample injection was made with a flow rate of 0.2 mL/min at 25 °C.

Detection was performed using an Agilent 6545 LC/Q-TOF quadrupole time-of-flight liquid mass spectrometry system. Ion detection employed single reaction ion monitoring, with electrospray ionization as the ionization mode. Secondary tandem mass spectrometry was conducted using a positive ion scan (*m*/*z*: 75–2000), with a capillary voltage of 2.0 kV, an ion source temperature of 200 °C, and a collision energy of 35.0 eV.

### 2.3. Cell Recovery and Culture

The H9c2 cells were obtained from the American Tissue Culture Collection (ATCC) of the USA and removed from the liquid nitrogen and quickly placed in a 37 °C water bath. They were then shaken rapidly for 1–2 min to ensure thorough thawing. The cell suspension was then transferred to a centrifuge tube. The cells were diluted with 5 mL of DMEM (Dulbecco’s Modified Eagle’s Medium, Thermo Scientific, Waltham, MA, USA) containing 10% fetal bovine serum (FBS) and mixed thoroughly. After centrifugation at 1500 rpm for 5 min, the supernatant was discarded, and fresh medium was added. The cells were inoculated at a density of 6 × 10^4^ cells/mL in 25 mL culture flasks and incubated at 5% CO_2_ and 37 °C.

### 2.4. Establishment of an H9c2 Cell Oxidatively Damaged Model and Cell Viability Analysis

H9c2 cells were seeded in 96-well microplates at a concentration of 5 × 10^3^ cells/well. Various concentrations of H_2_O_2_ (0, 20, 50, 100, 200, 400, and 600 mmol/L) were then added to the cells for 2 h. Cell viability was assessed using the CCK-8 assay. For the ACN treatment group, different concentrations of anthocyanins (50, 100, 200, 500, 1000, 1500, and 2000 μg/mL) were added to the cells for 6 h after H_2_O_2_ treatment, and cell viability was determined using the CCK-8 assay.

### 2.5. qPCR

H9c2 cells from each group were collected and digested with 0.25% trypsin. Total RNA was extracted using TRIzol reagent. The cDNA was synthesized by reverse transcription of total RNA (3 μg) using the iScript cDNA Synthesis kit (Takara, Beijing, China). RT-qPCR was performed using the CFX Connect Real-Time System (Bio-Rad) with the SYBR Green PrimeScript RT kit (TaKaRa) following the manufacturer’s instructions.

### 2.6. Western Blot

After culturing, 150 µL of cell lysis buffer was added to each well of a 6-well plate for 30 min. For rat tissue, 1 mL of lysis buffer was added per gram of tissue, followed by homogenization using a homogenizer. The supernatant was extracted by centrifugation at 10,000× *g* for 15 min at 4 °C to obtain the protein. The concentration of the protein sample was determined using the bicinchoninic acid (BCA) kit to quantify the sample amount. After running a 12% SDS–polyacrylamide gel electrophoresis (with a sample size of 20 µg, using voltages of 80 V and 100 V), the proteins were transferred from the gel to a nitrocellulose membrane (PVDF) (Bio-Rad, Hercules, CA, USA) for 90 min at 350 mA. After blocking at 25 °C for 1 h, the nitrocellulose membrane was incubated overnight at 4 °C with a primary antibody diluted in blocking buffer. It was rinsed with detergent three times, each time for 10 min. The secondary antibody was incubated for 1 h at room temperature. After rinsing three times for 10 min each, the enhanced chemiluminescence (ECL) method was used for detection. Western blot data should be analyzed using Quantity One29.0 for gray value statistics.

### 2.7. Apoptosis Assay of H9c2 Cells

H9c2 cells (5 × 10^4^) from various experimental groups were trypsinized, centrifuged at 1000× *g* for 5 min, and washed with phosphate-buffered saline (PBS). After removing the supernatant, cells were sequentially treated with 195 μL of Annexin V-FITC conjugate, 5 μL of Annexin V-FITC, and 10 μL of propidium iodide staining solution. The mixture was then incubated at room temperature (20–25 °C) in the dark for 10–20 min, followed by placement in an ice bath shielded from light with aluminum foil. Apoptosis rates were determined via flow cytometric analysis.

### 2.8. Transcriptome Information Analysis of H9c2 Cells

Total RNA was extracted from H9c2 cells and subjected to PCR enrichment to generate the final cDNA library. Clusters were formed on the cBot using the TruSeq PE Cluster Kit v3-cBot-HS (Illumina, San Diego, CA, USA) reagent. Subsequently, a double-end sequencing program (PE150) was used on the Illumina sequencing platform to generate 150 bp paired-end sequencing reads.

Species transcript expression was determined by aligning the sequencing reads to a reference genome, which enabled the identification of specifically expressed sequences for functional gene analysis.

### 2.9. Grouping and Administration of Rats with Myocardial Infarction

A total of 104 SPF Wistar rats, aged 8–10 weeks and equally divided between males and females, with an average body weight of 180 ± 20 g, were freely provided with food and water. They were housed under controlled conditions with the room temperature maintained at 22 ± 2 °C, humidity at 60 ± 5%, and a 12 h light–dark cycle. After a week of adaptation to the environment and diet, the rats with myocardial infarction were grouped and administered as follows:

Sham group: rats underwent the surgical procedure described for myocardial infarction; iSham group: rats underwent the surgical procedure described for myocardial infarction induction, but without ligation of the left coronary artery.

Myocardial infarction (MI) group: rats underwent the surgical procedure described for myocardial infarction induction, with ligation of the left coronary artery.

Treatment groups: MI rats were further divided into subgroups based on the treatment they received:

ACN high-dose group: received ACN at a dosage of 100 mg/kg.

ACN medium-dose group: received ACN at a dosage of 50 mg/kg.

ACN low-dose group: received ACN at a dosage of 25 mg/kg.

ACN administration was performed three times daily (at 9:00, 12:00, and 17:00) for a total duration of 6 weeks.

The myocardial infarction model was established as follows: Following intraperitoneal anesthesia with a combination of anesthetics (3 mL/kg), an incision was made on the left anterior chest to expose the heart by propping open the ribs. A 6-0 suture was threaded through the origin of the left coronary artery’s anterior descending branch, positioned between the inferior border of the left auricle and the pulmonary artery, along with ligating a small bundle of myocardium. Post ligation, the color of the ventricular wall and heartbeat were monitored using electrocardiography for 10 min. Upon complete cessation of bleeding, the chest cavity was closed, and the incision was sutured layer by layer. Approximately 1 h post operation, the rats gradually regained consciousness and returned to their normal state by the following day.

### 2.10. Histopathological Examination of Rats’ Heart Muscle Tissue

The hearts of the rats were carefully excised and promptly fixed in 4% paraformaldehyde in PBS treated with DEPC (diethylpyrocarbonate). Subsequently, the samples were dehydrated and then embedded in paraffin. The paraffin blocks were then continuously sectioned into slices with a thickness of 3–4 µm. Each case yielded 6–7 paraffin slices, which were subsequently deparaffinized in xylene and subjected to decreasing ethanol gradients (75%, 85%, 90%, 95%, and 100%), and finally washed with double-distilled water. Following deparaffinization and rehydration, the sections were stained with hematoxylin and eosin (H&E) dye.

### 2.11. Determination of NO, Nt-proBNP, Hs-CRP, and Angiotensin II Levels in Rats’ Heart Muscle Tissue

ROS and NO levels were measured following the method outlined by Choi et al. [20], with minor modifications. Additionally, experiments were conducted in accordance with the protocols provided with the ROS and NO assay kits(Njjcbio, Nanjing, China). Nt-proBNP levels were assessed using the methods described by Zhu et al. [21]. Hs-CRP and Angiotensin II levels were determined using the ELISA method as outlined by Zhang et al. [22].

### 2.12. Statistical Analysis

The data are presented as mean ± standard deviation (SD). Statistical analysis was conducted using analysis of variance (one-way ANOVA) followed by Duncan’s multiple range tests. Differences were considered statistically significant at *p* < 0.05. SPSS statistical software (version 13.0, SPSS Inc., Chicago, IL, USA) was used for the statistical analyses. Additionally, the Origin Pro 8.6 program (OriginLab Inc., Northampton, MA, USA) and GraphPad Prism 6.0 software were used for further statistical analyses and graphical representation, respectively.

## 3. Results

### 3.1. Purification and Identification of ACN

The samples were scanned with a UV-Vis spectrophotometer in the wavelength range of 400–600 nm. The anthocyanin standard is shown in Figure 2A. ACN exhibited absorption peaks in the visible region ranging from 500 to 550 nm, with the most prominent peak at 542 nm, aligning with the typical absorption peak of anthocyanins in the visible region (Figure 2B). Based on the results of HPLC-MS identification (Table 1 and Figure 2C–G), this study concluded that black highland barley is a rich source of six distinct anthocyanins. These are specifically Delphinidin-3-pyranoside, Malvidin-3-pyranoside, Delphinidin-3-glucoside, Petunidin-3-pyranoside, Cyanidin-3-glucoside, and Malvidin-3-acetylglucoside. The presence of these bioactive compounds in black highland barley contributes to its potential health benefits, particularly in the realm of cardiovascular health and antioxidant defense. Specifically, Delphinidin-3-pyranoside, Malvide-3-pyranoside, and Cyanidin-3-glucoside were found to be the predominant anthocyanins, constituting a significant 70% of the total anthocyanin content.

### 3.2. ACN Inhibits Oxidative Stress and Oxidative Stress-Mediated Apoptosis in H9c2 Cells

To investigate whether the purified anthocyanin (ACN) extracted from black barley possesses antioxidant properties that could help H9c2 cells resist damage induced by oxidative stress, we utilized a hydrogen peroxide (H_2_O_2_)-induced cellular oxidative stress model, supplemented with ACN. Subsequently, the antioxidant effects of ACN were validated using CCK-8 and DCFH-DA staining, as shown in Figure 3A–C. The results demonstrated a significant decrease in cell viability (*p* < 0.0001) and a notable increase in intracellular reactive oxygen species (ROS) levels (*p* < 0.001) in the H_2_O_2_-treated group. Conversely, the group supplemented with ACN exhibited significantly enhanced cell viability and reduced intracellular levels (*p* < 0.05). These findings suggest that ACN can effectively reduce oxidative stress-induced cellular damage in vitro.

Cellular oxidative stress damage is often associated with apoptosis. Therefore, we assessed the anti-apoptotic capability of ACN in the H9c2 cellular oxidative stress model using flow cytometry, as depicted in Figure 3D,E. We observed that ACN significantly reduced H9c2 cell apoptosis induced by hydrogen peroxide in a dose-dependent manner. Given the mitochondria-related nature of apoptosis, we investigated the mitochondrial status in cells from each group using transmission electron microscopy (Figure 3D). The results revealed that oxidative stress induced by hydrogen peroxide led to mitochondrial vacuolization in H9c2 cells, with abnormal mitochondria accounting for over 60% of the total mitochondrial population. Supplementation with ACN significantly decreased the proportion of abnormal mitochondria in cells (Figure 3F). Furthermore, we evaluated the expression of apoptosis-related proteins within cells, including BAX, Bcl-2, and cleaved caspase-3 (Figure 3H–J). We found that ACN supplementation dose-dependently reduced the levels of the apoptosis proteins BAX and cleaved caspase-3 induced by oxidative stress while increasing the expression of the anti-apoptotic protein Bcl-2. These results collectively indicate that ACN can alleviate hydrogen peroxide-mediated oxidative stress damage in H9c2 cells, improve mitochondrial morphology, and inhibit apoptosis.

### 3.3. RNA-seq Reveals That the Antioxidant Effect of ACN Operates through the PTEN-Akt Pathway

Through the aforementioned experiments, we observed that ACN exhibits significant antioxidative effects on H9c2 cells. However, the molecular mechanisms underlying its actions remain unclear. RNA-seq can be used to explore the potential molecular mechanisms of drugs. Therefore, we divided H9c2 cells into four groups for mRNA-seq analysis: the control group (CK), the hydrogen peroxide-treated group (H_2_O_2_), the anthocyanin-treated group (ACN), and the group treated with anthocyanin combined with hydrogen peroxide (ACN + H_2_O_2_). The gene expression levels of each group are illustrated in Figure 4A. Subsequently, hierarchical clustering analysis was conducted for each group, as shown in Figure 4B, indicating alterations in the expression patterns of genes across groups. Following this, we conducted KEGG functional annotation and pathway analysis on these differentially expressed genes and generated metabolic pathway diagrams as illustrated in Figure 4C,D. In these diagrams, red indicates upregulated genes or proteins, green indicates downregulated genes or proteins, blue represents mapped genes or proteins, and white and purple denote background colors. Through the metabolic pathway diagrams, we observed a significant upregulation of the PTEN gene in cells treated with hydrogen peroxide, which was attenuated by anthocyanin treatment.

Initially, we validated the sequencing results using qPCR, as shown in Figure 4E, con firming the reduction in PTEN mRNA levels following ACN treatment. The PTEN/Akt signaling pathway has been extensively studied, with PTEN being recognized for its role in inhibiting the phosphorylation of Akt (*p*-Akt). p-Akt, in turn, regulates apoptosis by inhibiting the activity of transcription factors such as FKHR, NF-κB, and YAP, while promoting the activity of CREB and Mdm2, among others [23]. Furthermore, p-Akt directly inhibits the phosphorylation of proteins such as Bad, GSK-3, and caspase-9, thereby exerting anti-apoptotic effects. Based on this, we hypothesize that the mechanism by which ACN inhibits apoptosis may be mediated through the PTEN/Akt pathway. To investigate this further, we assessed the phosphorylation status of Akt kinase in cells using Western Blot analysis. The results depicted in Figure 4F demonstrate that the addition of ACN enhances the reduction in p-Akt induced by H_2_O_2_, thereby exerting its anti-apoptotic effects.

### 3.4. ACN Accelerates the Recovery of Myocardial Cells in a Mouse Model of Myocardial Infarction

Although our in vitro experiments clearly demonstrate the antioxidant effects of anthocyanin, further investigation is needed to understand its effects in vivo. To achieve this goal, we established a rat model of myocardial infarction. After administering anthocyanin to the model rats for a period, we evaluated myocardial cell death using HE staining, as shown in Figure 5A. In the sham-operated control group, the myocardial morphology appeared normal, with a clear structure and orderly arrangement of myocardial fibers. In the myocardial infarction group, a significant reduction in myocardial cells was observed in the infarct area. This reduction was accompanied by nuclear fragmentation, disappearance, and replacement by disordered connective tissue. Additionally, there was an infiltration of numerous granulocytes and monocytes. Following treatment with anthocyanin at various concentrations, we observed a significant increase in the number of myocardial cells and a gradual restoration of the orderly arrangement of myocardial fibers, resembling the tissue morphology of the sham-operated group.

Subsequently, we measured the expression of Nt-proBNP, NO, Hs-CRP, and Angiotensin II in myocardial cells of each experimental group using ELISA to assess the recovery of myocardial cells after myocardial infarction, as shown in Figure 5B–E. The results indicated that the elevation of Nt-proBNP, Hs-CRP, and Angiotensin II levels observed in the myocardial infarction experimental group was inhibited by anthocyanin treatment in a dose-dependent manner. Additionally, the production of nitric oxide (NO) significantly increased after anthocyanin treatment.

Furthermore, we evaluated the effect of ACN on the infarct area of mouse hearts, as depicted in Figure 5F,G. The results indicated a significant reduction in the infarct area of model mice as the concentrations of anthocyanin increased. These results collectively indicate that ACN improves the recovery of myocardial injury after acute myocardial infarction. In the previous mRNA-seq results, it was discovered that anthocyanin may help in resisting oxidative stress-induced cell apoptosis through the PTEN/Akt/Bcl-2 pathway. To investigate this further, we collected myocardial tissues from each treatment group and performed Western blot analysis. As shown in Figure 5H, the expression of the apoptotic protein caspase-3 significantly decreased after anthocyanin treatment, and the activation state of the PTEN-Akt pathway under oxidative stress conditions was also significantly inhibited. These results suggest that ACN plays a positive role in ameliorating myocardial damage after acute myocardial infarction, and this effect is mediated through the PTEN-Akt pathway.

## 4. Discussion

Over the coming decades, with aging populations and increasing rates of obesity and diabetes, the burden and healthcare costs associated with cardiovascular disease (CVD) are expected to rise significantly on a global scale. Despite extensive efforts to elucidate the pathophysiological mechanisms underlying the occurrence and progression of cardiovascular disease, much work remains to be done. Current research suggests that intracellular oxidative stress is a major contributor to the pathogenesis of the disease [24]. Given the prevalence of cardiovascular disease and the role of oxidative stress in various cardiovascular conditions, there is widespread interest in the development of natural antioxidants to alleviate or prevent cardiovascular disease.

Anthocyanins are natural water-soluble pigments widely present in plants, renowned for their potent antioxidant properties, surpassing early discoveries such as butylated hydroxyanisole (BHA), butylated hydroxytoluene (BHT), and α-tocopherol [25,26]. In recent years, numerous studies have demonstrated the diverse biological activities of anthocyanins, including antioxidant, anti-inflammatory, anticancer, and anti-apoptotic activities [27]. Meanwhile, the potential mechanisms were gradually revealed. However, there is relatively limited research on their protective effects against cardiovascular diseases. In this study, we provide evidence that anthocyanins exhibit antioxidant and anti-apoptosis effects both in vitro and in vivo, conferring protective effects against cardiovascular diseases.

Oxidative stress injury is a common form of cellular damage that results from an imbalance between the production and clearance of reactive oxygen species (ROS). While many biological processes rely on low concentrations of ROS under physiological conditions, excessive ROS production can lead to pathological conditions. Many studies have shown that excessive ROS production can cause severe damage to myocardial cells, disrupting the balance of the oxidative–antioxidative system, thereby inducing mitochondrial abnormalities and triggering cell apoptosis [28]. In a previous study, blueberry anthocyanin extract and anthocyanin-rich extract of sour cherry exhibited decreased ROS generation and an increased gene expression of NOS, HO-1, and SOD [29,30]. A recent study demonstrated that anthocyanin derived from berries exerted cytoprotective effects against H_2_O_2_-induced oxidative stress in diabetic aortic endothelial cells by attenuating cytotoxicity induced by H_2_O_2_ [31]. In this study, we also found that treatment with ACN from black highland barley reduced the levels of ROS in oxidative stress, decreased the proportion of abnormal mitochondria, and inhibited cell apoptosis (see Figure 3A–G). Therefore, inhibiting oxidative stress-induced damage and cell apoptosis is a crucial intervention strategy for cardiovascular diseases.

Apoptosis plays a crucial part in diabetic CVD etiologies. However, anthocyanin is a well-known bioactive compound in ameliorating diabetes mellitus that stimulates apoptosis in CVD. This has been confirmed by a study that revealed that anthocyanin treatment resulted in a higher-level response of anti-apoptotic Bcl-2 and a lower-level response of proapoptotic caspase-3 and BAX [32]. Anthocyanin from black highland barley also reduces caspase-3 and BAX levels and increases Bcl-2 levels, which further attenuate apoptosis (see Figure 3H and Figure 5H).

For further research on the potential mechanisms of anthocyanin-modulating oxidative stress injury and apoptosis, RNA sequencing (RNA-seq) serves as a powerful tool for delving into the underlying molecular mechanisms of anthocyanin. In our study, we meticulously stratified H9c2 cells into four distinct experimental groups to conduct a comprehensive mRNA-seq analysis. By examining the metabolic pathway diagrams, we discerned a pronounced overexpression of the PTEN gene in cells exposed to hydrogen peroxide. Notably, this overexpression was effectively mitigated by subsequent treatment with anthocyanin (see Figure 4E,F). This observation suggests that anthocyanin may hold potential therapeutic value by modulating the expression of key regulatory genes such as PTEN, thereby influencing cellular responses to oxidative stress.

Phosphatase and tensin homolog (PTEN) belong to the protein tyrosine phosphatase (PTP) family and was initially identified as a tumor suppressor, playing specific roles in regulating cell growth. Among other PTPs, PTEN’s unique phosphatase function involves dephosphorylating 3,4,5-triphosphorylated phosphoinositides (PIP3) to generate PIP2, thereby counteracting the activity of phosphatidylinositol 3-kinase (PI3K) [33]. Through this mechanism, PTEN acts as an inhibitor of the phosphatidylinositol 3-kinase/protein kinase B (PI3K/Akt) pathway. Several studies have investigated the PI3K/PTEN/Akt signaling pathway, with PTEN recognized for its role in inhibiting Akt phosphorylation. Phosphorylated Akt (p-Akt) regulates apoptosis by inhibiting the activity of transcription factors such as FKHR, NF-κB, and YAP, promoting CREB and Mdm2 and directly inhibiting the phosphorylation of proteins such as Bad, GSK-3, and caspase-9, thereby exerting anti-apoptotic effects [23]. Lee et al. first demonstrated the reversible inactivation of PTEN by H_2_O_2_. In this process, the catalytic residue Cys124 of PTEN’s active site is oxidized and forms a disulfide bond with Cys71, leading to inactivation. This inactivation is reversible because oxidized PTEN is reduced back to its active form by the oxidative–reductive system, particularly the thioredoxin (Trx) system commonly present in the cellular environment [34]. Huang and his colleagues have demonstrated that administrations of blueberry anthocyanin extract, malvidin, malvidin-3-glucoside, and malvidin-3-galactoside were able to significantly ameliorate injured endothelial cells by elevating endogenous SOD and lowering ROS production. The same study also showed that anthocyanin effectively induced vasodilatory effects by increasing nitric oxide (NO) and its promoter, endothelial NO synthase (eNOS), as well as via the activation of the phosphoinositide 3-kinase (PI3K)/Akt signaling pathway [29].

In our investigation, we found that mRNA levels of the PTEN protein significantly increased in H9c2 cells stimulated by hydrogen peroxide through mRNA-seq, and its protein levels also increased significantly, as confirmed by Western blot analysis. However, the addition of ACN downregulated the upregulation of PTEN protein induced by H_2_O_2_ stimulation (see Figure 4E,F and Figure 5H). This suggests that ACN may regulate oxidative stress-induced myocardial cell apoptosis by modulating the expression level of PTEN. Further investigation revealed that ACN’s promotion of PTEN expression ultimately led to an increase in p-Akt levels, thereby activating downstream pathways to inhibit the expression of cleaved-caspase 3 and consequently suppressing apoptosis. These results suggest that ACN may inhibit oxidative stress-induced myocardial cell apoptosis by regulating PTEN expression to alter Akt signaling transduction just like the metabolic pathway diagrams induced by mRNA sequencing (Figure 4C,D).

Natural products serve as important sources for novel therapeutic agents. At present, the clinical utilization of antioxidants for the prevention and treatment of cardiovascular diseases is somewhat limited. In our investigation, we have provided evidence, through both in vitro and in vivo experiments, that ACN exerts a protective influence against myocardial cell apoptosis triggered by oxidative stress. Delving into the underlying mechanisms, our research has revealed that the PTEN and Akt signaling pathways are instrumental in the apoptosis of myocardial cells under oxidative stress conditions. ACN effectively mitigates apoptosis by enhancing PTEN expression, thereby resulting in a subsequent reduction in the levels of phosphorylated Akt (p-Akt). Moreover, it is important to note that natural products frequently engage multiple molecular targets and are implicated in intricate signaling cascades.

mRNA-seq revealed that ACN significantly altered PTEN expression levels, thus elucidating a partial role of ACN in oxidative stress-induced myocardial cell apoptosis. However, mRNA-seq also revealed that ACN altered other pathways, suggesting that ACN may exert its anti-apoptotic effects through other mechanisms as well. Furthermore, in this study, we did not investigate how ACN alters PTEN expression levels. Our future research will focus on exploring how ACN modulates the expression levels of PTEN. We hope that this study will provide a solid foundation for the clinical application of ACN in cardiovascular diseases. Additionally, the present study did not address the mechanisms by which ACN influences PTEN expression levels. Our forthcoming research endeavors will be concentrated on elucidating the regulatory effects of ACN on PTEN expression. It is our expectation that this line of inquiry will pave the way for a more comprehensive understanding of ACN’s potential clinical utility in the treatment of cardiovascular diseases.

## 5. Conclusions

In summary, we found that anthocyanin from black highland barley notably reduced H9c2 cell damage caused by oxidative stress and effectively inhibited apoptosis by activating the PTEN-Akt signaling pathway. Meanwhile, this study reminded us that natural compounds, such as ACN, possibly interact with multiple molecular targets and engage in intricate signaling cascades. It is of great significance to the development of foods or drugs for the prevention and treatment of cardiovascular diseases.

## Figures and Tables

**Figure 1 foods-13-01417-f001:**
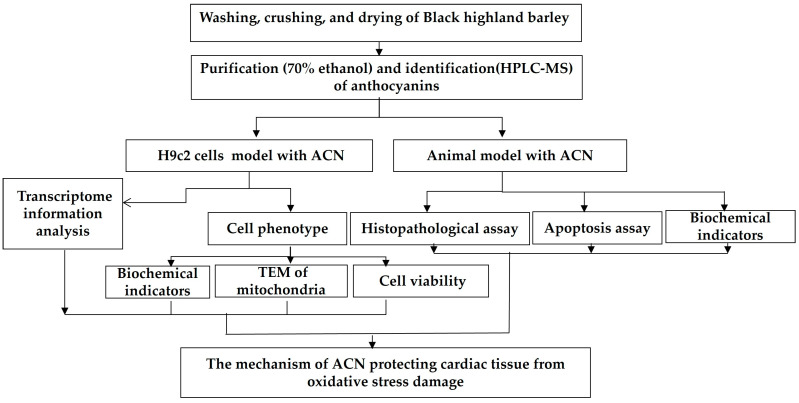
Schematic diagram of experimental procedure.

**Figure 2 foods-13-01417-f002:**
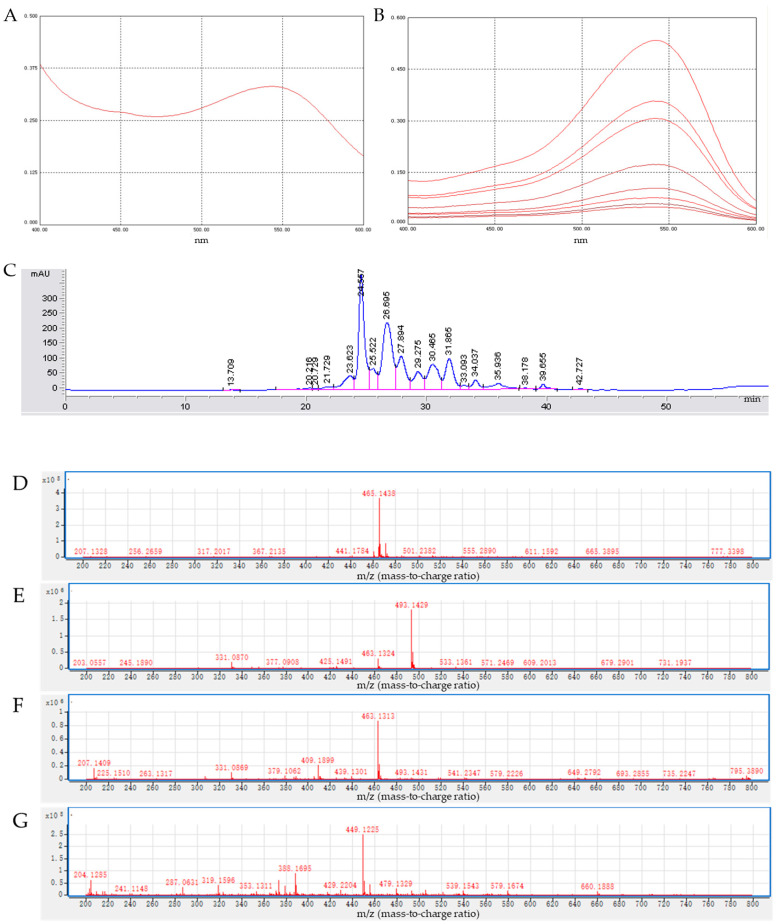
Purification and identification of anthocyanin in black highland barley. (**A**) UV–visible scan spectrum of anthocyanin standard sample. (**B**) UV–visible scan spectrum of anthocyanins in Black highland barley. (**C**) RP-HPLC chromatogram of anthocyanins in black highland barley. (**D**,**E**) LC-MS/MS analysis and identification of anthocyanins in black highland barley. (**D**) Delphinidin-3-pyranoside. (**E**) Malvidin-3-pyranoside. (**F**) Delphinidin-3-glucoside. (**G**) Cyanidin-3-glucoside and Petunidin-3-pyranoside.

**Figure 3 foods-13-01417-f003:**
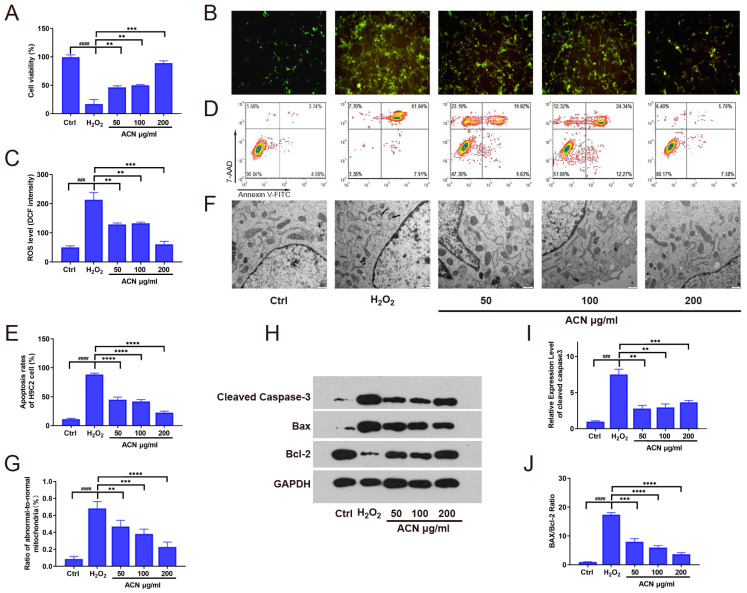
ACN suppressed oxidative stress and oxidative stress-induced apoptosis in H9c2 cells. H9c2 cells were initially treated with H_2_O_2_ and subsequently exposed to different concentrations of ACN (0.625, 1.25, and 2.5 µM). (**A**) Cell viability assessed by CCK8 assay. (**B**,**C**) Intracellular ROS levels. (**D**,**E**) Flow cytometry analysis of apoptosis. (**F**,**G**) Transmission electron microscopy images of cellular mitochondria, showing the proportion of abnormal mitochondria to total mitochondria. (**H**–**J**) Expression levels of cleaved caspase-3, Bax, and Bcl-2 proteins in cells, as well as the ratio of BAX/Bcl-2 expression level. **: *p* < 0.01, *** (###): *p* < 0.001, **** (####): *p* < 0.0001. *p*-values for differences between cohorts in mean scores in tests are based on analysis of variance (one-way ANOVA).

**Figure 4 foods-13-01417-f004:**
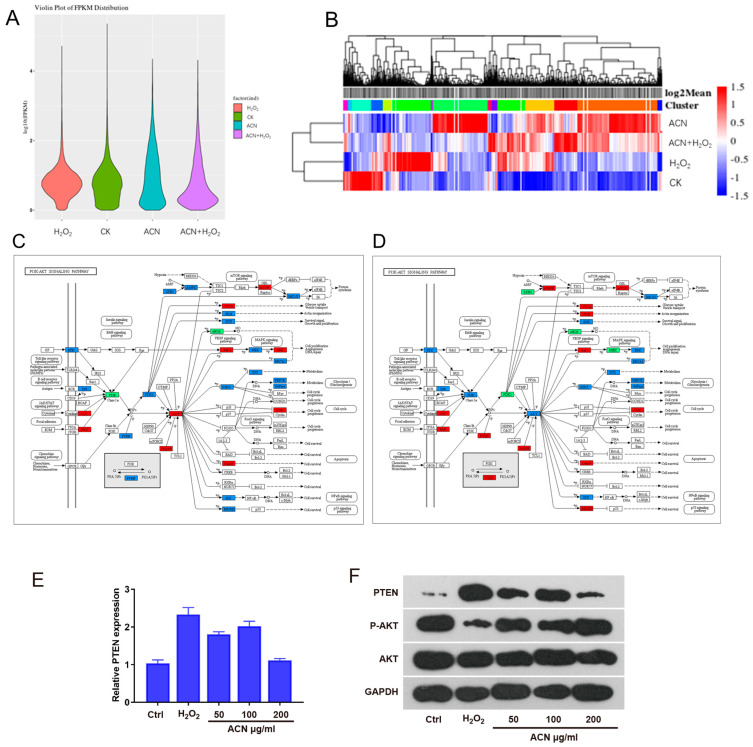
RNA-Seq results of H9c2 cells treated with H_2_O_2_, CK, ACN, and ACN + H_2_O_2_ groups. (**A**) Box plots illustrating the distribution of FPKM values. (**B**) Hierarchical clustering analysis of significantly differentially expressed genes among groups. (**C**,**D**) Metabolic pathway diagrams of H9c2 cells treated with H_2_O_2_ (**C**) and ACN + H_2_O_2_ (**D**). (**E**) mRNA levels of PTEN protein in cells of each group. (**F**) Expression levels of PTEN-Akt pathway-related proteins in cells of each group.

**Figure 5 foods-13-01417-f005:**
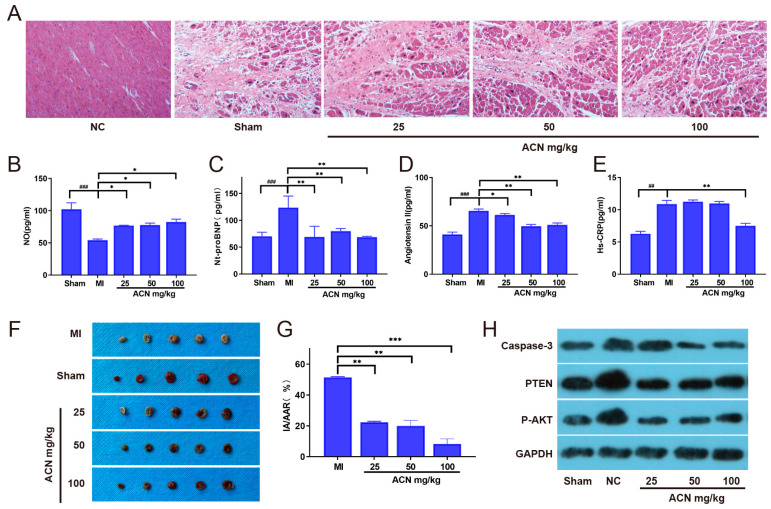
ACN aids in cell recovery in a mouse model of myocardial infarction (MI). (**A**) HE staining of myocardial tissues in each group. (**B**–**E**) Measurement of Nt-proBNP, NO, Hs-CRP, and Angiotensin II levels in myocardial cells of each experimental group using ELISA. (**F**,**G**) Effects of ACN on myocardial infarct area in MI model mice. (**H**) Expression levels of PTEN-Akt pathway-related proteins in MI model mice of each group. *: *p* < 0.1, ** (##): *p* < 0.01, *** (###): *p* < 0.001. *p*-values for differences between cohorts in mean scores in tests are based on analysis of variance (one-way ANOVA).

**Table 1 foods-13-01417-t001:** The composition of anthocyanins in black highland barley.

Peak No.	Name	Retention Time (min)	*m*/*z* (Precursor)	*m*/*z* (Product Ion)	Identified
1	Delphinidin-3-pyranoside	24.567	465.1	303	Yes
2	Malvide-3-pyranoside	26.695	493.1	331	Yes
3	Delphinidin-3-glucoside	27.895	463.1	331	Yes
4	Petunidin-3-pyranoside	29.275	479.1	317	Yes
5	Cyanidin-3-glucoside	30.465	449.1	287	Yes
6	Malvidin-3-acetylglucoside	31.865	535.0	331	Yes

## Data Availability

The original contributions presented in the study are included in the article, further inquiries can be directed to the corresponding author.

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
