# Peer review of "Anthocyanin of Black Highland Barley Alleviates H2O2-Induced Cardiomyocyte Injury and Myocardial Infarction via Activating the Phosphatase and Tensin Homolog/Phosphatidylinositol 3-Kinase/Protein Kinase B Pathway"

_foods, 2024, doi:10.3390/foods13091417_

Round 1

Reviewer 1 Report

Comments and Suggestions for Authors

The introduction provides a comprehensive overview of the global significance of cardiovascular disease (CVD) and the potential benefits of anthocyanidins. However, it could be strengthened by delving deeper into the specific mechanisms by which anthocyanidins may contribute to cardiovascular health and also some of the methodologies and tests used should also be explained here, to introduce the second part of the article, conveniently.

Additionally, explicitly stating the objectives of the study and outlining the research questions or hypotheses in the introduction would provide readers with a clearer understanding of the study's purpose.

In the discussion section, the main findings are effectively summarized and contextualized within the broader literature on CVD and anthocyanidins. However, to enhance the discussion, it would be valuable to include references to other relevant studies and to compare and contrast the findings of this study with those of previous research. This would help readers better understand the significance of the current study within the existing body of literature and identify any consistencies or discrepancies in the findings.

Considering the breadth of topics covered, it may be beneficial to divide the discussion component content into two separate works.

Overall, by incorporating these suggestions, the paper could provide a more comprehensive analysis of the relationship between anthocyanidins and cardiovascular health, contributing to the broader understanding of this important topic.

Author Response

The introduction provides a comprehensive overview of the global significance of cardiovascular disease (CVD) and the potential benefits of anthocyanidins. However, it could be strengthened by delving deeper into the specific mechanisms by which anthocyanidins may contribute to cardiovascular health and also some of the methodologies and tests used should also be explained here, to introduce the second part of the article, conveniently.

“the specific mechanisms by which anthocyanidins may contribute to cardiovascular health” ——This has been added to the second paragraph of the introduction.

Additionally, explicitly stating the objectives of the study and outlining the research questions or hypotheses in the introduction would provide readers with a clearer understanding of the study's purpose.

——Please refer to the fifth paragraph of the introduction.

In the discussion section, the main findings are effectively summarized and contextualized within the broader literature on CVD and anthocyanidins. However, to enhance the discussion, it would be valuable to include references to other relevant studies and to compare and contrast the findings of this study with those of previous research. This would help readers better understand the significance of the current study within the existing body of literature and identify any consistencies or discrepancies in the findings.

——In the discussion section, references are added and supplemented to compare and contrast the findings of this study.

Considering the breadth of topics covered, it may be beneficial to divide the discussion component content into two separate works.

——The content of the discussion has been divided into several levels for discussion. Additionally, a fifth section of conclusions has been added.

Reviewer 2 Report

Comments and Suggestions for Authors

This study used various chemical analytic techniques to quantify the anthocyanin content in highland barley and evaluated their antioxidant capacities through stress-induced damage in myocardial cells and uncover the molecular mechanism at play. The results indicate that highland barley is a good source of bioactive compounds with health-promoting properties.

I would like to address the points as follows:

1.       The introduction requires supplementing with the characteristics of the raw material used. The authors briefly describe anthocyanins, but there is no detailed description of the plant material in terms of its use and health-promoting properties.

2.       The study did not mention the standardisation of extraction and analysis methods for polyphenolic compounds. Quantification lacks methodological examinations that include linearity. Standardisation is important to ensure the accuracy and reproducibility of results.

3.       Why did only the authors focus on the analysis of anthocyanins? Numerous reports show that black barley is also a source of other classes of polyphenolic compounds that have strong antioxidant properties and, in this experiment, could also have a significant impact on the results obtained.

4.       Figure 1 section D-G the presented spectra are illegible, I suggest magnification

5.       Table 1 I suggests standardising the names of the identified compounds. What type of pyranosides have been identified. I suggest sticking to one regimen, glucoside or glucopyranoside. Peak no. 3, error in m/z for Delphinidin aglycone, it should be 303.

6.       I suggest placing the summary in a separate conclusion paragraph.

Author Response

This study used various chemical analytic techniques to quantify the anthocyanin content in highland barley and evaluated their antioxidant capacities through stress-induced damage in myocardial cells and uncover the molecular mechanism at play. The results indicate that highland barley is a good source of bioactive compounds with health-promoting properties.

I would like to address the points as follows:

  1. The introduction requires supplementing with the characteristics of the raw material used. The authors briefly describe anthocyanins, but there is no detailed description of the plant material in terms of its use and health-promoting properties.

——Please refer to the paragraph 3 of the introduction.

  1. The study did not mention the standardisation of extraction and analysis methods for polyphenolic compounds. Quantification lacks methodological examinations that include linearity. Standardisation is important to ensure the accuracy and reproducibility of results.

——Given the variety of plant materials and extracted components, a uniform extraction protocol for anthocyanins and polyphenols appears to be elusive. In the case of anthocyanins from black highland barley, we have detailed the extraction and identification procedures in our publication titled "Isolation, Purification, and Identification of Anthocyanins from Black Highland Barley".

  1. Why did only the authors focus on the analysis of anthocyanins? Numerous reports show that black barley is also a source of other classes of polyphenolic compounds that have strong antioxidant properties and, in this experiment, could also have a significant impact on the results obtained.

——One contributing factor is the comparatively high anthocyanin content found in black highland barley among various crops. Additionally, the significance of anthocyanins' protective role against cardiovascular diseases has gained increasing recognition in recent years.

  1. Figure 1 section D-G the presented spectra are illegible, I suggest magnification

——Revised

  1. Table 1 I suggests standardising the names of the identified compounds. What type of pyranosides have been identified. I suggest sticking to one regimen, glucoside or glucopyranoside. Peak no. 3, error in m/z for Delphinidin aglycone, it should be 303.

——“Delphinidin-3-pyranoside, Malvidin-3-pyranoside, Delphinidin-3-glucoside, Pe-tunidin-3-pyranoside, Cyanidin-3-glucoside, and Malvidin-3-acetylglucoside”are standard names. m/z for Delphinidin aglycone has been corrected. Thanks.

  1. I suggest placing the summary in a separate conclusion paragraph。

——Please refer to“5. Conclusion”.

Reviewer 3 Report

Comments and Suggestions for Authors

The authors explored the cardioprotective properties of anthocyanin (ACN), a compound derived from black barley, against oxidative stress-induced damage in myocardial cells to uncover the molecular mechanisms, and involved in vitro and in vivo experimental models. The findings appeared very interesting, however, there are areas of the work that require further improvements, as below:
a) Please, in the introduction, create a new paragraph (best to make this paragraph 3) about black barley, how it is broken down, about the various bioactive components that emerge from it, and examples of various methods that have being employed to extract them (make sure that the extraction method used in this current work is among the examples)
b) Kindly start the materials and methods with a new subsection captioned 'Schematic overview of the experimental program", which should comprise of about 4 sentences, and must be accompanied by a flow diagram. Please, this flow diagram should show the entire process of the entire methodology, how the samples were collected, processed, extraction, and how sampling was allocated to various tested parameters. This is to help guide the readers. Please, make sure to connect the flow diagram contents with the objective of this work. This will help readers see the link between the objective of this work with the methodology

c) results is well presented. Because results and discussion are separated, please, make best effort to ensure all figures and table in the results are captured in the discussion and use "(Refer to Fig. x)" or "(Refer to Table 1)" in all the places where the data from specific Table or Fig is used. Now, if there are aspects that have not been fully captured, this is an opportunity to check through the discussion. Please make sure that all the data from each Figure and Table in results are captured in the discussion

d) Please, create a conclusion section, the last paragraph of the discussion..so that it stands out on its own.

Look forward to your revised manuscript.

Author Response

The authors explored the cardioprotective properties of anthocyanin (ACN), a compound derived from black barley, against oxidative stress-induced damage in myocardial cells to uncover the molecular mechanisms, and involved in vitro and in vivo experimental models. The findings appeared very interesting, however, there are areas of the work that require further improvements, as below:

  1. a) Please, in the introduction, create a new paragraph (best to make this paragraph 3) about black barley, how it is broken down, about the various bioactive components that emerge from it, and examples of various methods that have being employed to extract them (make sure that the extraction method used in this current work is among the examples)

——Please see the paragraph 3 in the introduction.

  1. b) Kindly start the materials and methods with a new subsection captioned 'Schematic overview of the experimental program", which should comprise of about 4 sentences, and must be accompanied by a flow diagram. Please, this flow diagram should show the entire process of the entire methodology, how the samples were collected, processed, extraction, and how sampling was allocated to various tested parameters. This is to help guide the readers. Please, make sure to connect the flow diagram contents with the objective of this work. This will help readers see the link between the objective of this work with the methodology

——Please see “2.1 Overview of experimental design and procedures”.

  1. c) results is well presented. Because results and discussion are separated, please, make best effort to ensure all figures and table in the results are captured in the discussion and use "(Refer to Fig. x)" or "(Refer to Table 1)" in all the places where the data from specific Table or Fig is used. Now, if there are aspects that have not been fully captured, this is an opportunity to check through the discussion. Please make sure that all the data from each Figure and Table in results are captured in the discussion

——Revised

  1. d) Please, create a conclusion section, the last paragraph of the discussion.so that it stands out on its own.

——Please refer to “5. Conclusion”.

Round 2

Reviewer 2 Report

Comments and Suggestions for Authors

The authors mostly responded positively to my comments. However, I still believe that issues regarding the standardization and validation of the anthocyanin quantification method should be clarified. If authors refer to earlier work, this should be noted in the manuscript.

Author Response

We illustrate the problem of standardization and validation of anthocyanin quantification methods based on literature data from previous studies.

  1. Liu, ,Li, W., Hu, Z., Qin, X., Liu. G. (2020). Isolation, purification, identification, and stability of anthocyanins from Lycium ruthenicum Murr. LWT-Food Science and Technology,126, Article 109334. doi: 10.1016/j.lwt.2020.109334.
  2. Barnes,S., Nguyen,H.P., Shen, S., Schug, K.A. (2009). General method for extraction of blueberry anthocyanins and identification using high performance liquid chromatography-electrospray ionization-ion trap-time of flight-mass spectrometry. Chromatogram. A, 1216 (23): 4728-4735. doi: 10.1016/j.chroma.2009.04.032
